# A countercurrent microflow strategy for simultaneous high selectivity and conversion in aromatic nitration

Jing Song, Yongqi Pan, Ruobing Xin, Zifei Yan [ID], Tianyao Tang, Kai Wang [ID], Yujun Wang, Jian Deng & Guangsheng Luo [ID] [✉]

Aromatic nitration, a hazardously complex process, poses serious risks. A major challenge for the reaction is the trade-off effect between spatiotemporal conversion rate and selectivity, particularly the over-nitration side reactions that have plagued the field for nearly 200 years. We propose a countercurrent microflow mode between two microreactors, which boosts spatiotemporal conversion rate by over five times compared to the normal single-stage co-current microflow mode, and two orders of magnitude compared to traditional batch reactors. Meanwhile, we identify an inhibition mechanism of over-nitration. The generated $H_2O$ in the main reaction can in situ reduce the dissolution of nitroaromatics in the aqueous phase and effectively prevent over-nitration. Through synergistic control of both kinetics and thermodynamics in the microreaction process, high spatiotemporal conversion and selectivity are achieved simultaneously, overcoming the trade-off effect. Furthermore, we demonstrate the broad applicability of the microflow strategy across various aromatic nitration processes.

Aromatic nitration is one of the top well-known chemical reactions, with nearly 200 years of history. It is widely used in the production of pharmaceuticals, dyes, pesticides, and energetic materials. Due to its low cost and high nitrating activity, the mixed acid composed of concentrated $H_2SO_4$ and $HNO_3$ is the most widely used nitrating agent, and is almost the only choice of nitrating agent in the aromatic nitration industry[1]. The aromatic nitration with mixed acid is a typical fast and highly exothermic reaction, with the reaction medium being highly corrosive and posing significant risk of explosion[2]. As such, it is regarded as a classic hazardous chemical process. From the 1830s to the 1990s, aromatic nitration was predominantly carried out in stirred-tank reactors. Due to the limited transport rates of stirred tanks, the process was typically operated under low-temperature and low-reactant-concentration conditions to ensure safety. Since the 1990s, there has been a gradual shift toward the use of microreactors and flow chemistry for aromatic nitration. Microreactors offer superior heat and mass transport performance compared to stirred tanks and have a

small liquid holdup, providing a higher level of safety[3–5]. However, the hazards of aromatic nitration not only exist in reaction process, but also arise during the storage, transportation, and separation of reaction products. This is primarily due to the side reactions in aromatic nitration that produce high-energy and thermally unstable byproducts. The two most prominent side reactions are over-nitration and oxidation. It has been shown that oxidation side reactions can be effectively suppressed in microreactors[6]. Nonetheless, over-nitration side reaction remains inevitable due to the complex parallel-consecutive reactions (Fig. 1a)[7].

In stirred tanks, a relatively high mononitration selectivity could be achieved at the cost of significantly reduced spatiotemporal conversion rate[8–10]. In microreactors, the spatiotemporal conversion rate is greatly improved[6,11–13]. However, all existing microreaction technologies remain insufficient to control the selectivity of aromatic nitration. The intense heat release of the reaction causes unavoidable rises in interfacial temperatures, which significantly exacerbate over-

State Key Laboratory of Chemical Engineering and Low-Carbon Technology, Department of Chemical Engineering, Tsinghua University, Beijing, China.
[✉]e-mail: gsluo@tsinghua.edu.cn

a. Parallel–consecutive reactions

b. Electrophilic substitution reaction mechanism

(1)    *Protonation reaction:*    $HNO_3 + H^+ \longrightarrow NO_2^+ + H_2O$

(2)

**Fig. 1 | Reaction network and mechanism of aromatic nitration with mixed acid. a** Reaction network. **b** Reaction mechanism.

nitration, resulting in notably lower mononitration selectivity in microreactors compared to stirred tanks. Taking the toluene nitration as an example (Supplementary Fig. 1), the selectivity for mono-nitrotoluene ($S_{MNT}$) in stirred-tank reactors can exceed 95%, while in current microreactors, $S_{MNT}$ ranges between 64% and 95%[14–16]. The characteristics result in a pronounced trade-off effect between spatiotemporal conversion rate and selectivity even in the microreaction systems. Furthermore, based on the reaction mechanism of electrophilic substitution, the more active the nitration reaction is, the more significant the trade-off effect is (Fig. 1b)[17–19]. Unfortunately, no effective solution to this issue has been identified over the past two centuries. In addition, due to the kinetic characteristics of a second-order reaction, the low reactant concentrations in the later reaction stages lead to significantly reduced intrinsic reaction rates[3,20,21]. Current methods for achieving high aromatic conversion involve either extending the residence time or increasing the HNO$_3$ feed. However, due to the characteristics of consecutive reactions, extended residence times exacerbate over-nitration and lead to high-pressure drops in flow chemistry systems. Similarly, increasing HNO$_3$ feed intensifies competitive side reactions, further intensifying the over-nitration issue (Fig. 1a). To overcome these bottlenecks in aromatic nitration, it is imperative to develop new strategies that enhance spatiotemporal conversion rate without compromising selectivity.

Here, based on the principles of flow chemistry, we proposed to introduce the concept of countercurrent flow into flow chemistry to address the significant disparity in reaction rate and interfacial temperature distribution during the early and late stages under co-current conditions. A two-stage countercurrent kinetic control strategy featuring co-current flow intra-stage, countercurrent flow inter-stage is proposed. Based on the improved controllability of the interfacial temperature, the thermodynamic properties of the nitration system were further characterized, and the regulatory mechanism of interfacial thermodynamic property changes on reaction selectivity during the reaction was explored. Finally, by synergistically integrating kinetic and thermodynamic control, we achieve both a high spatiotemporal conversion rate and high selectivity in aromatic nitration simultaneously. The method is also extended to the nitration of other typical aromatics, such as benzene and chlorobenzene, to demonstrate the broad applicability of the proposed strategy.

## Results

### Comparison of reaction kinetics between co-current and countercurrent modes

According to the second-order kinetics, the relationship between aromatic conversion rate and residence time in a conventional single-stage co-current flow mode can be described as Eq. (1) (derivation in

Supplementary Note 1)[22].

$$x_{ar} = \frac{\frac{c_N^0}{c_{ar}^0}(1 - e^{k_1(c_N^0 - c_{ar}^0)t})}{1 - \frac{c_N^0}{c_{ar}^0}e^{k_1(c_N^0 - c_{ar}^0)t}} \tag{1}$$

where $x$ is the aromatic conversion rate, $c$ is the concentration, $k_1$ is the observed reaction rate constant for mononitration and $t$ is the residence time. Superscript 0 denotes the initial condition, and subscripts ar and N refer to the aromatic and HNO$_3$, respectively. We assume isothermal conditions. The generated H$_2$O has no effect on the reaction rate, as well as ignoring the over-nitration. Thus, $k_1$ is constant.

Based on Eq. (1), Fig. 2a shows that over 90% conversion occurs within the first 10% of residence time due to high reactant concentrations. As the reaction progresses, the decreasing concentrations substantially slow down the reaction rate. In practice, H$_2$SO$_4$ dilution by the generated H$_2$O reduces $k_1$ over time, making actual residence times longer than predicted. This is an inevitable result of the second-order reactions under co-current flow. However, the over-nitration, as the second step in the consecutive reactions, will be exacerbated when longer residence times are applied to achieve higher aromatic conversion, resulting in a reduced selectivity for mononitration, as shown in Fig. 2b.

To address this, we propose a countercurrent microflow mode of co-current flow intra-stage, countercurrent flow inter-stage by combining the idea of countercurrent flow and flow chemistry (Fig. 2c). This mode transforms the conventional single-stage co-current system into two co-current stages. Aromatic feed enters the first-stage and reacts with the mixed acid derived from the second-stage. Due to the excess of aromatic, HNO$_3$ in the aqueous phase could be completely consumed, yielding an organic phase consisting unreacted aromatic and nitroaromatic products. This stream, combined with fresh mixed acid, enters the second-stage, where the remaining aromatic is fully nitrated. The mixed acid stream is recycled back to the first-stage, completing the countercurrent flow. In this configuration, aromatic conversion in each stage affects the reactant concentrations in the other, thereby influencing overall reaction efficiency (Fig. 2d). As the aromatic conversion in the first-stage ($x_{ar,1}$) increases, the initial HNO$_3$ concentration in the first-stage rises, enhancing its reaction rate. According to the mass conservation, this reduces the initial aromatic concentration in the second-stage, lowering its reaction rate. Consequently, the total reaction efficiency, as the sum of both stages, shows a nonlinear relationship with $x_{ar,1}$, as further discussed in the next section. The residence times required in both modes are compared under identical conditions (Fig. 2d). For instance, to achieve 99.9% conversion of aromatic, the two-stage countercurrent demonstrates

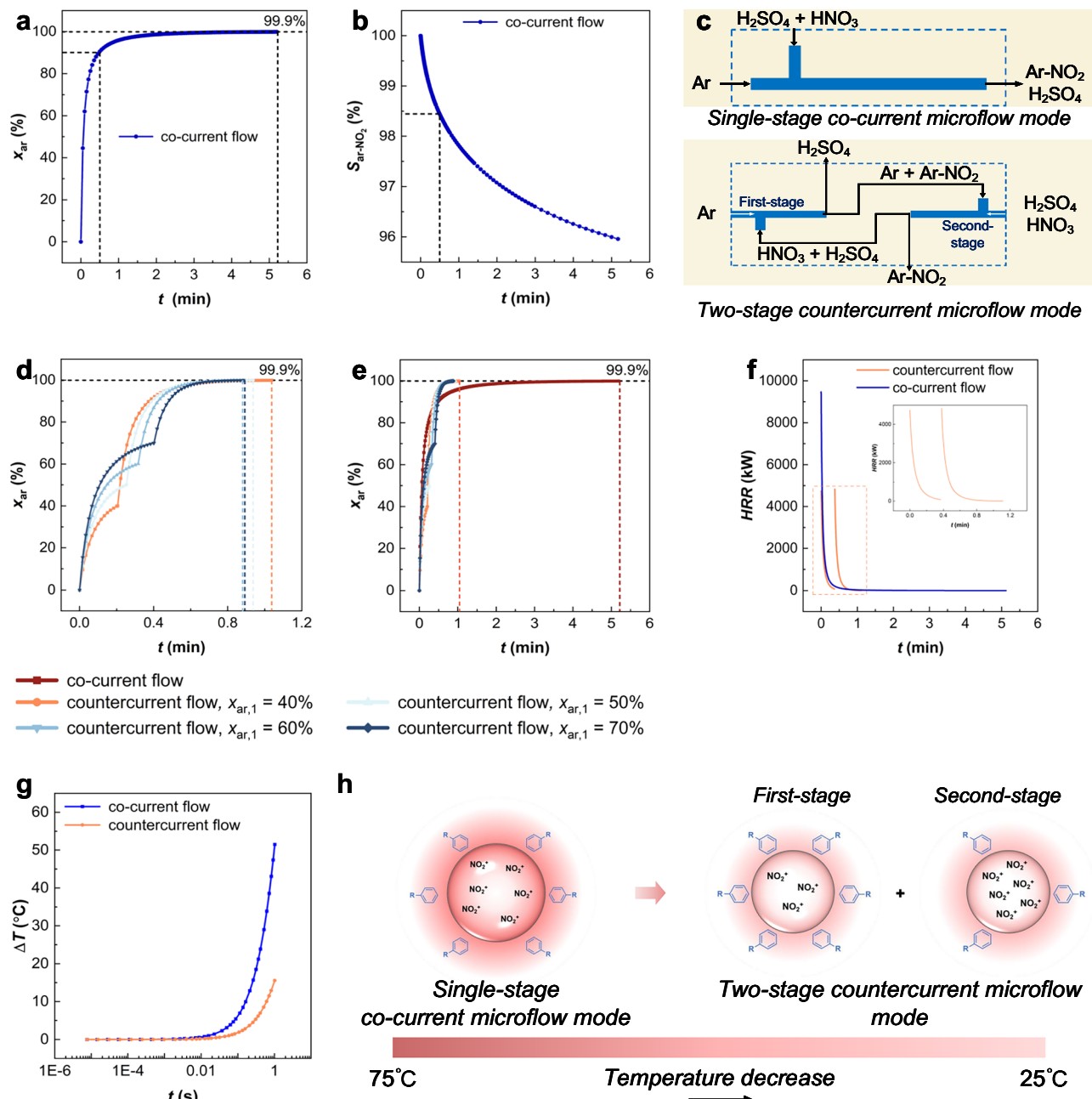

**Fig. 2 | Comparison of reaction performance of aromatic nitration in single-stage co-current and two-stage countercurrent modes. a** Variation of aromatic conversion ($x_{ar}$) with residence time ($t$) in the single-stage co-current microflow. **b** Variation of nitroaromatic selectivity ($S_{ar-NO_2}$) with $t$ in the single-stage co-current microflow. **c** Schematic of single-stage co-current and two-stage countercurrent modes. **d** Effect of aromatic conversion in the first-stage ($x_{ar,1}$) on total reaction efficiency in the two-stage countercurrent mode. **e** Comparison of reaction efficiency. **f** Comparison of heat release rate (*HRR*). **g** Comparison of interface temperature rise ($\Delta T$). **h** Schematic of interface temperature and system composition, as quantified in (**g**). (Assume that reaction rate constant $k_1 = 0.05$ L/(mol s), initial aromatic concentration $c_{ar} = 5$ mol/L, initial HNO₃ concentration $c_N = 5.05$ mol/L). Source data are provided as a Source data file.

an over 5 times improvement in reaction efficiency (Fig. 2e). This efficiency gap widens with increasing target conversion. Furthermore, lower reactant concentrations in each stage of the two-stage countercurrent mode greatly reduce the heat release rate (*HRR*) compared to the single-stage co-current mode (Fig. 2f). The maximum of *HRR* in the countercurrent mode is only half that of the single-stage co-current mode, significantly reducing the difficulty of controlling the system interface temperature. A comparison of interfacial temperature evolution during the early stage of the reaction is shown in Fig. 2g. Under the co-current mode, the interfacial temperature rises by 51 °C within

1 s, whereas in the two-stage countercurrent mode, the temperature increases by only 15 °C, representing a 69.8% reduction.

In summary, the two-stage countercurrent mode significantly enhances reaction efficiency and facilitates the achievement of a high conversion rate, while simultaneously reducing the heat release rate of the system and improving the controllability of the aromatic nitration process. It should be noted that the two-stage countercurrent mode does not change the intrinsic reaction rate constant and activation energy, but rather redistributes the concentration gradients of the two reactants along the reactor, enabling each stage to operate under more

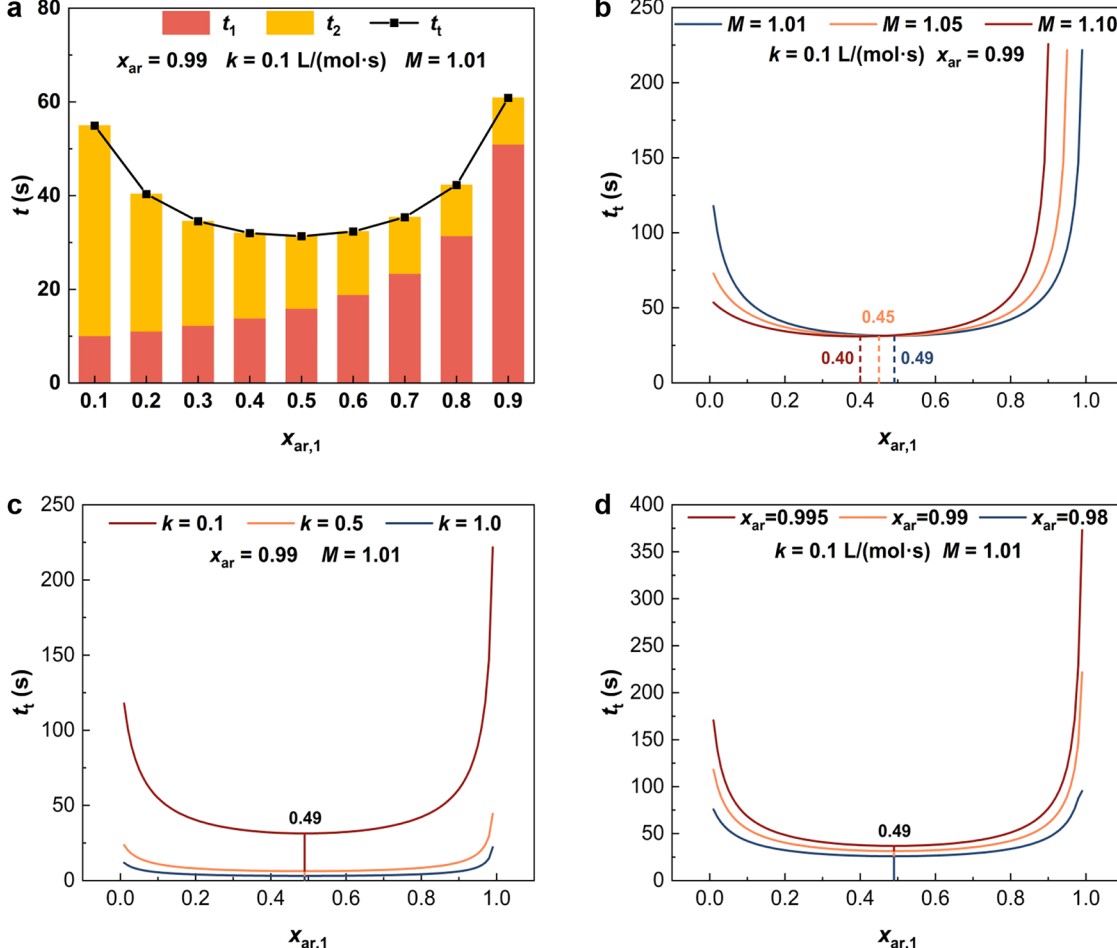

**Fig. 3 | Effect of operating parameters on total residence time ($t_t$) in the two-stage countercurrent mode. a** Effect of single-stage aromatic conversion ($x_{ar,1}$) on the residence time of each stage and total residence time, where $t_1$ and $t_2$ denote the residence time in the first-stage and second-stage microreactor, respectively. **b** Effect of initial molar ratio ($M$) of $HNO_3$ to aromatic on total residence time. **c** Effect of apparent reaction rate constant ($k$) on total residence time. **d** Effect of target conversion ($x_{ar}$) on total residence time. Source data are provided as a Source data file.

favorable local conditions and thus enhancing overall spatiotemporal conversion rate and thermal uniformity.

## Optimization of reaction efficiency of two-stage countercurrent mode

Each stage in the two-stage countercurrent mode operates under co-current microflow. Therefore, the total residence time required to achieve a target overall conversion is the sum of the residence times needed for each stage to achieve its respective conversion. Based on the second-order reaction kinetics, this relationship is expressed in Eq. (2) (derivation in Supplementary Note 2).

$$t_t = \frac{1}{k[c_N^0 \cdot c_{ar}^0(1 - x_{ar,1}) \cdot c_{ar}^0]} \ln\left[\frac{c_{ar}^0(1 - x_{ar})}{c_{ar}^0 - [c_N^0 \cdot c_{ar}^0(1 - x_{ar,1})]x_{ar}}\right]$$
$$+ \frac{1}{k[c_{ar}^0(1 - x_{ar,1}) \cdot c_N^0]} \ln\left[\frac{c_N^0(1 - x_{ar})}{c_N^0 \cdot c_{ar}^0(1 - x_{ar,1})x_{ar}}\right] \quad (2)$$

where $t_t$ is the total residence time. $x_{ar}$ is the total target aromatic conversion. $x_{ar,1}$ is the aromatic conversion achieved in the first-stage. The variation of $t_t$ with $x_{ar,1}$ is illustrated in Fig. 3a. As $x_{ar,1}$ increases, the residence time in the first-stage ($t_1$) increases, while that in the second-stage ($t_2$) decreases. $t_t$ initially decreases and then increases with the increase of $x_{ar,1}$. This indicates the existence of an optimal first-stage

conversion (($x_{ar,1}$)$_{opt}$), at which $t_t$ is minimized. The effects of the reaction rate constant ($k$), the initial molar ratio of $HNO_3$ to aromatic ($M$), and the overall target conversion ($x_{ar}$) on ($x_{ar,1}$)$_{opt}$ are further investigated (Fig. 3b–d). The results show that ($x_{ar,1}$)$_{opt}$ decreases with the increase in $M$. However, when $M$ is fixed, $k$ and $x_{ar}$ only affect the values of $t_t$ but do not change ($x_{ar,1}$)$_{opt}$. These findings suggest that optimizing $x_{ar,1}$ is an effective method to minimize $t_t$, when $M$ changes. ($x_{ar,1}$)$_{opt}$ can be determined by Eq. (3).

$$(x_{ar,1})_{opt} = 0.5 - (M - 1) = 1.5 - M \quad (3)$$

This provides an effective method for optimizing the efficiency of the two-stage countercurrent mode for second-order reactions. Given a specific $M$, the respective aromatic conversion rate allocated to the two stages can be determined.

The kinetic regulation method described above is based on the principles of flow chemistry and established from the principles of second-order reaction kinetics. Therefore, this method is broadly applicable beyond aromatic nitration and can be extended to other second-order reaction systems.

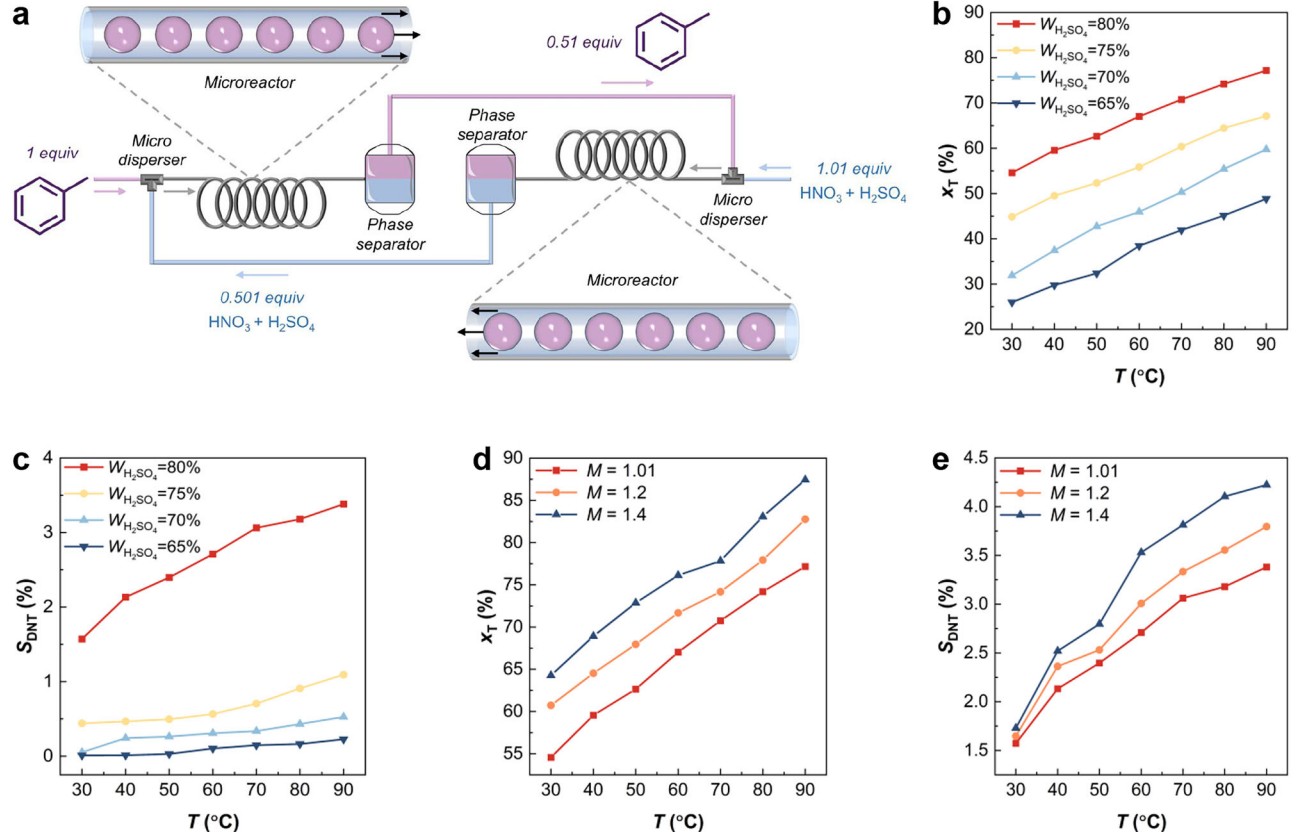

**Fig. 4 | Process design and performance control of toluene nitration in the two-stage countercurrent microreactor. a** Schematic of the molar equivalent of substances in the two-stage countercurrent nitration of toluene. **b** Effects of temperature ($T$) and $H_2SO_4$ concentration ($W_{H2SO4}$) on toluene conversion ($x_T$). **c** Effects of $T$ and $W_{H2SO4}$ on the selectivity toward dinitrotoluene ($S_{DNT}$). ($M = 1.01$, $t_t = 1.96$ min, $Q_a = Q_o = 0.5$ mL/min, $D_i = 0.5$ mm) **d** Effect of initial molar ratio ($M$) of $HNO_3$ to aromatic on $x_T$. **e** Effect of $M$ on $S_{DNT}$. ($W_{H2SO4} = 80\%$, $t_t = 1.96$ min, $Q_a = Q_o = 0.5$ mL/min, $D_i = 0.5$ mm). Here, $D_i$ is the diameter of microreactor. $Q_a$ and $Q_o$ are the volumetric flow rates of aqueous and organic phase, respectively. Source data are provided as a Source data file.

## Application of two-stage countercurrent kinetic regulation

The kinetic regulation strategy was applied to aromatic nitration, using toluene nitration as a representative example (Supplementary Note 3). As mentioned above, the enhancement of the spatiotemporal conversion rate originates from the optimized reactant concentration profiles governed by the law of mass action. The molar ratio of $HNO_3$ to aromatic ($M$) serves as the key parameter that regulates the concentration distribution of the two reactants along the reactor. It also determines the allocation of conversion between the two stages in the design of the two-stage countercurrent mode, as described by Eq. (3).

Unlike the single-stage co-current mode, which requires a large excess of $HNO_3$ to achieve high toluene conversion, the two-stage countercurrent mode enables high toluene conversion with stoichiometric feeding, assuming over-nitration is negligible. However, in practice, a slight excess of $HNO_3$ is necessary to account for its consumption via over-nitration side reactions and to help suppress equipment corrosion. Therefore, this study adopts $M = 1.01$, representing a 1% excess of $HNO_3$. According to Eq. (3), $(x_{ar,1})_{opt}$ is equal to 0.49. This indicates that the overall reaction efficiency is maximized when the first-stage conversion reaches 49%. Given the target overall conversion of 99.9%, the corresponding molar equivalent design of the microreaction system is illustrated in Fig. 4a.

## Suppress over-nitration side reaction via thermodynamic regulation

Aromatic nitration suffers from the trade-off effect between spatiotemporal conversion rate and selectivity. The two-stage

countercurrent mode significantly improves the spatiotemporal conversion rate, making suppression of over-nitration reaction essential to overcoming trade-off effect. Temperature ($T$), $H_2SO_4$ concentration ($w_{H2SO4}$), and $M$ are critical parameters influencing over-nitration. As shown in Fig. 4b–e, increasing $T$, $w_{H2SO4}$, or $M$ accelerates the reaction but also exacerbates the over-nitration due to the increased interfacial temperatures, reflecting the trade-off effect. Notably, the selectivity of over-nitration ($S_{DNT}$) is highly sensitive to $w_{H2SO4}$, and decreases greatly when $w_{H2SO4}$ falls below 80% (Fig. 4c). Under the operating conditions explored in this work, $S_{DNT}$ is much lower than values reported in the literature (5–36%), where higher $H_2SO_4$ dosages are commonly employed.

$H_2SO_4$ serves as both catalyst and solvent, affecting both reaction kinetics and product distribution through physical solubility changes. Conventional aromatic nitration processes rely on high $H_2SO_4$ dosage to moderate temperature rise and ensure safety. In contrast, the superior transport property of microreactor allows for greatly reduced $H_2SO_4$ dosage while maintaining safety, enabling exploration of scientific blind spots in aromatic nitration. Based on the principles of flow chemistry, the $H_2SO_4$ dosage determines the volumetric flow rate ratio between the aqueous ($Q_a$) and organic phases ($Q_o$). The effect of $Q_a/Q_o$ on the reaction performance in the second-stage microreactor was examined (Fig. 5a, b and Supplementary Fig. S2).

Decreasing $H_2SO_4$ dosage lowers $Q_a/Q_o$ while increasing the initial $HNO_3$ concentration, resulting in a higher initial reaction rate when $Q_a/Q_o > 1$ (Fig. 5a). Meanwhile, $S_{DNT}$ decreases with decreasing $Q_a/Q_o$ (Fig. 5b). When $Q_a/Q_o < 1$, $S_{DNT}$ approaches zero, but the reaction rate

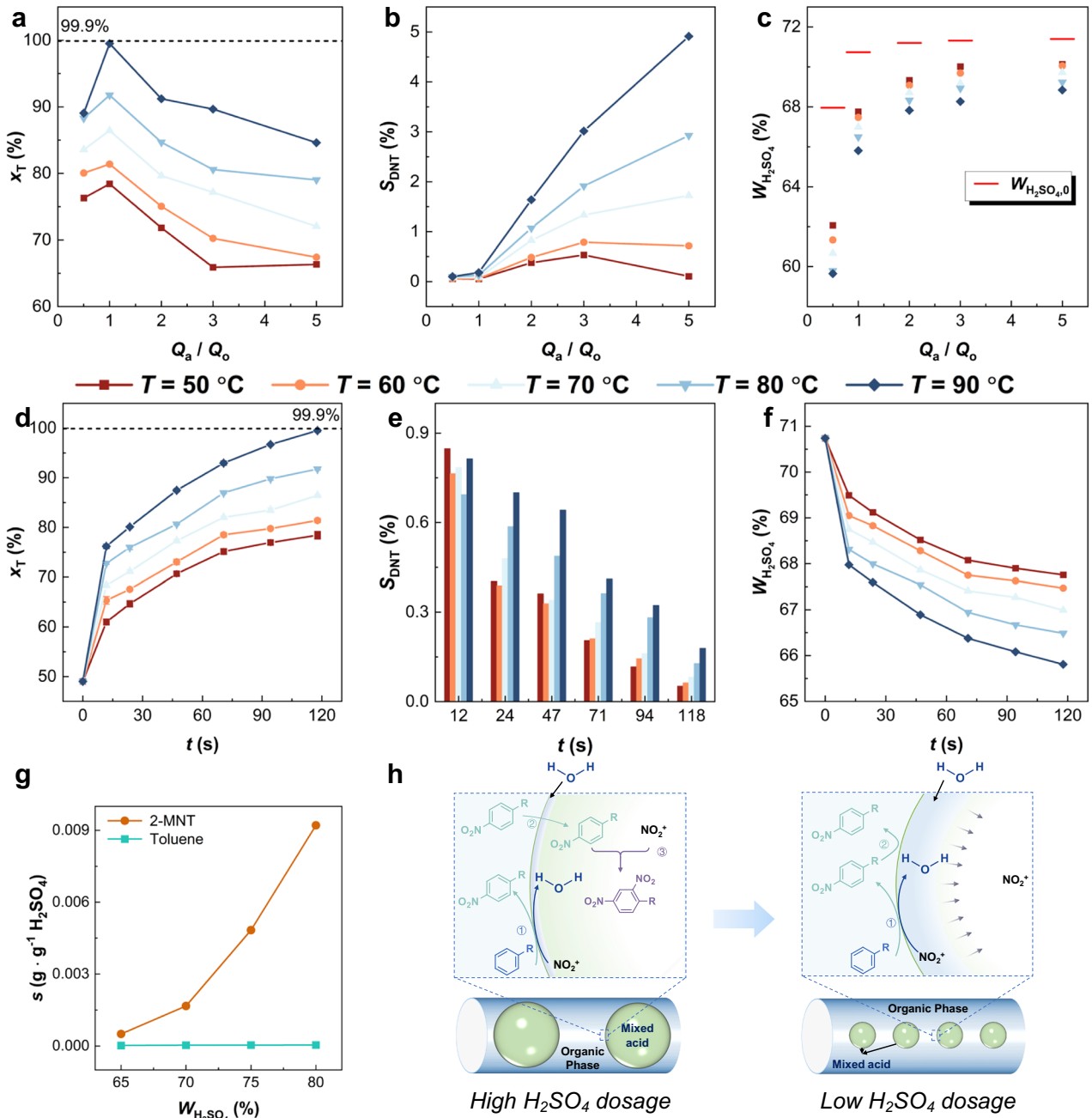

**Fig. 5 | Regulation of over-nitration side reaction selectivity of toluene nitration in the second-stage. a** Effects of temperature ($T$) and volumetric flow rate ratio ($Q_a/Q_o$) on toluene conversion ($x_T$). **b** Effects of $T$ and $Q_a/Q_o$ on the selectivity toward dinitrotoluene ($S_{DNT}$). **c** Changes in $H_2SO_4$ concentration ($W_{H2SO4}$) before and after the reaction ($M$ = 1.01:0.51, $t_t$ = 1.96 min, $D_i$ = 0.5 mm), the horizontal line represents the initial concentration of $H_2SO_4$ ($W_{H2SO4,0}$) in the aqueous phase, and the points represent the concentration of $H_2SO_4$ after the reaction. **d** Effects of $T$ and residence time ($t$) on $x_T$. **e** Effects of $T$ and $t_t$ on $S_{DNT}$. **f** Changes in $W_{H2SO4}$ before and after the reaction ($M$ = 1.01:0.51, $Q_a$ = $Q_o$ = 0.5 mL/min, $D_i$ = 0.5 mm). **g** Solubility ($s$) of toluene and $o$-nitrotoluene (2-MNT) at different $H_2SO_4$ concentrations. **h** Schematic of product inhibition mechanism. Source data are provided as a Source data file.

also decreases due to insufficient $H_2SO_4$ concentration to sustain $NO_2^+$ generation. Figure 5c shows that lower $H_2SO_4$ dosage enhances the dilution effect of generated $H_2O$, further reducing $H_2SO_4$ concentration and suppressing over-nitration. At $Q_a/Q_o$ = 1, both high conversion and high selectivity are achieved. This operating condition corresponds to an optimal dilution window (Supplementary Note 12), in which the molar ratio $n_{H2SO4}/n_{HNO3}$ is maintained between 0.7 and 1.5.

To clarify the role of $H_2O$, the effect of residence time on reaction performance and $H_2SO_4$ concentration was explored (Fig. 5d–f and

Supplementary Fig. S3). Increasing residence times decreased both reaction rate and $S_{DNT}$, indicating that over-nitration is more sensitive to $H_2O$ dilution than mononitration. The solubility of toluene and MNT in different $H_2SO_4$ concentrations was investigated (Fig. 5g). MNT is always more soluble than toluene in $H_2SO_4$, but as $H_2SO_4$ concentration decreases, the solubility of MNT decreases more significantly. These results imply that in conventional aromatic nitration processes (Fig. 5h), where $H_2SO_4$ dosage is large, the amount of generated $H_2O$ by the main reaction is low relative to the total $H_2SO_4$ dosage, resulting in minimal dilution effect. The interfacial temperature increases due to

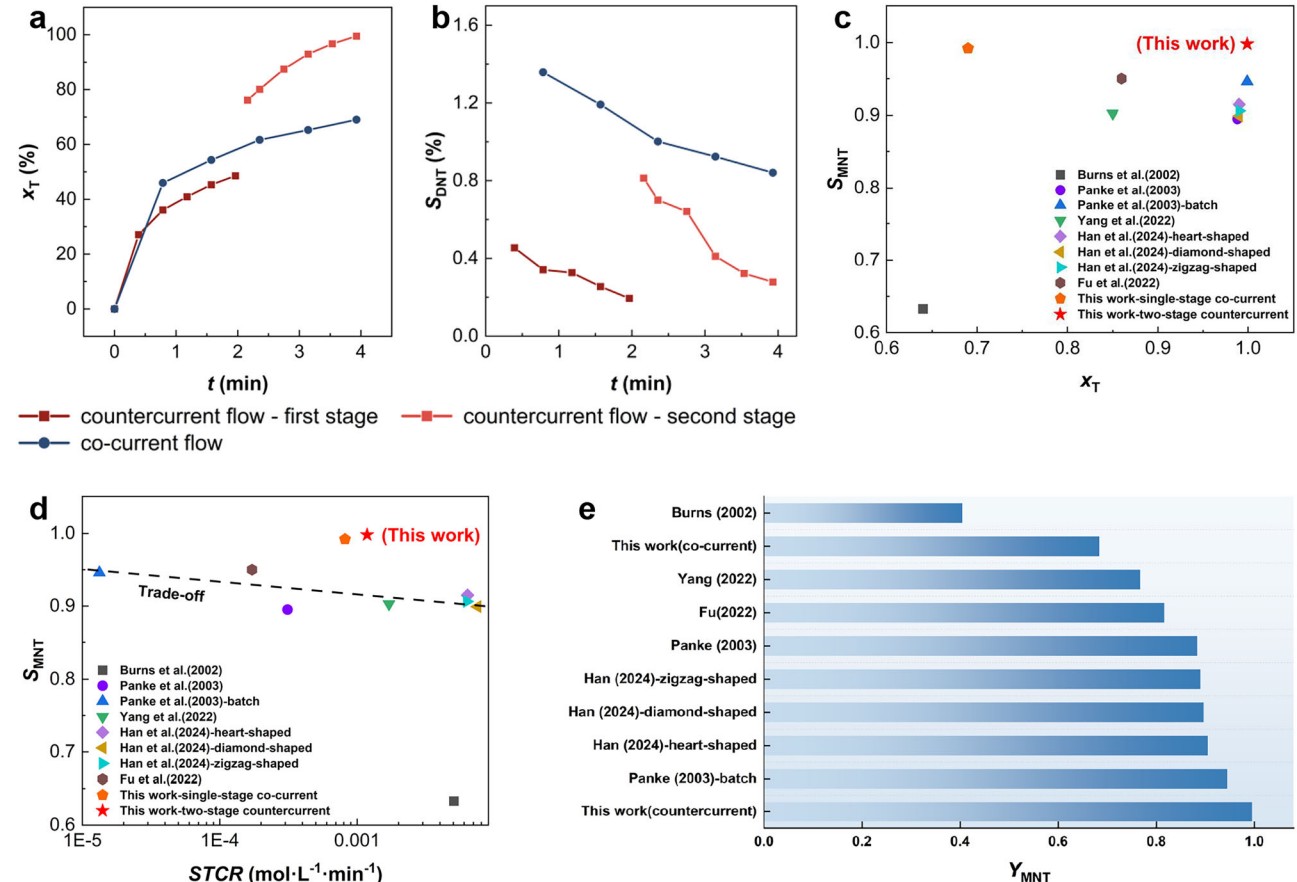

- ■ countercurrent flow - first stage    ■ countercurrent flow - second stage
- ● co-current flow

**Fig. 6 | Overcome trade-off effect in aromatic nitration via kinetics and thermodynamics regulation. a** Comparison of toluene (T) conversion ($x_T$) under single-stage co-current and two-stage countercurrent modes. **b** Comparison of dinitrotoluene (DNT) selectivity ($S_{DNT}$) under single-stage co-current and two-stage countercurrent modes ($W_{H2SO4,0} = 71.5\%$, $M = 1.01$, $Q_o = Q_a = 0.5$ mL/min, $T = 90$ °C, $t = 3.92$ min, $D_i = 0.5$ mm). **c** Comparison of $x_T$ and selectivity ($S_{MNT}$) of mononitrotoluene (MNT) in this work with those in the literatures. **d** Comparison of toluene spatiotemporal conversion rate ($STCR$) and $S_{MNT}$ in this work with those in the literatures. **e** Comparison of MNT yield ($Y_{MNT}$) in this work with those in the literatures[6,8,14–16]. Where $W_{H2SO4,0}$ is initial concentration of $H_2SO_4$. $M$ is initial molar ratio of $HNO_3$ to aromatic. $Q_a$ and $Q_o$ are the volumetric flow rates of aqueous and organic phase, respectively. Source data are provided as a Source data file.

the heat released by the reaction, resulting in a higher solubility of MNT in the mixed acid, where over-nitration occurs. In contrast, under low $H_2SO_4$ dosage, the generated $H_2O$ effectively in situ dilutes $H_2SO_4$, greatly reducing MNT solubility and preventing its transfer into the acid phase. Meanwhile, the higher initial $HNO_3$ concentration maintains the main reaction rate.

This mechanism enhances the reaction rate while simultaneously suppressing over-nitration side reactions in aromatic nitration. Its essence lies in the fact that the generated $H_2O$ in the main reaction in situ reduces the solubility of the product in $H_2SO_4$, thereby keeping the over-nitration side reaction under thermodynamically unfavorable conditions and effectively suppressing it. The concentration and dosage of $H_2SO_4$ are the key parameters for achieving this thermodynamic control, which we term the product inhibition mechanism (Fig. 5h).

Molecular simulations provide microscopic support for this mechanism (Supplementary Note 7). Nitroaromatic solvation in $H_2SO_4$ is dominated by directional hydrogen bond between the nitro group (-NO₂) and the -OH groups of $H_2SO_4$, whereas solvation in $H_2O$ is significantly weaker due to longer hydrogen bond distances. As $H_2O$ accumulates, the hydrogen bond network of the acid phase transitions from an $H_2SO_4$-dominated to an $H_2O$-dominated structure, markedly decreasing the solvating power of the acid phase. Furthermore, thermodynamic calculations reveal that increasing $H_2O$ content lowers the equilibrium concentration of nitronium ions ($NO_2^+$), thereby

kinetically suppressing over-nitration. Consequently, the droplet interface becomes a region where over-nitration is simultaneously thermodynamically and kinetically disfavored, representing a general principle applicable to aromatic nitration systems.

## Overcome trade-off effect in aromatic nitration via kinetics and thermodynamics regulation

The two-stage countercurrent mode serves as a reaction efficiency enhancement strategy based on kinetics, and the product inhibition mechanism offers a thermodynamic approach for selectivity control. Building on these principles, we developed a two-stage countercurrent microreaction system with $Q_a/Q_o = 1$ to emphasize the product inhibition effect, thereby enabling synergistic regulation through combined thermodynamic and kinetic control. Toluene nitration was first used as a model reaction to compare the performance of the two-stage countercurrent and single-stage co-current microreaction processes under identical operating conditions (Fig. 6a, b). With 1% excess $HNO_3$, the two-stage countercurrent achieved a 99.9% toluene conversion, while the single-stage co-current mode reached only 69%. The two-stage process also demonstrated lower over-nitration side reaction selectivity, which was only 0.2%, 1–2 orders of magnitude lower than the 5–36% reported in the literatures (Supplementary Notes 8, 10 and 11). The Hatta number (Ha) under the operating conditions was evaluated (Supplementary Note 9). The calculated Ha values in both stages of the microreactor are consistently below 0.3 and decrease

with increasing residence time, indicating that the reaction rate is governed by kinetics.

Through the synergistic control of kinetics and thermodynamics, we achieve high toluene conversion and MNT selectivity simultaneously (Fig. 6c). More importantly, the results reported in the literatures demonstrate a clear trade-off effect between spatiotemporal conversion rate and selectivity (Fig. 6d). Although relatively high conversion and selectivity were also achieved in batch reactors, this work improves the spatiotemporal conversion rate by two orders of magnitude (Supplementary Note 11), effectively overcoming the trade-off effect in nitration (Fig. 6d). Based on the product inhibition mechanism, the single-stage co-current also achieves high MNT selectivity, but it is difficult to reach a high conversion rate. As a result, the yield of MNT is significantly lower than that of the two-stage countercurrent (Fig. 6e). This limitation arises from the intrinsic coupling between reaction intensity, interfacial temperature rise, and residence time in the single-stage reaction mode.

We established a microflow strategy to overcome the trade-off effect based on the common laws of aromatic nitration kinetics and thermodynamics. To validate its universality, we extended it to the nitration reactions of benzene and chlorobenzene, with the reaction performance and solubility characteristics provided in Supplementary Notes 5 and 6. Due to the electron-donating effect of the methyl group, toluene exhibits stronger nitration reactivity, making the trade-off effect most prominent in toluene nitration. In contrast, the suppression of over-nitration side reaction in benzene and chlorobenzene is easier. Thus, applying the proposed method to the nitration of benzene and chlorobenzene yielded better results.

## Discussion

Aromatic nitration is fundamentally constrained by a trade-off between spatiotemporal conversion rate and selectivity, posing a serious challenge to its further development. Here, we developed a strategy to overcome the trade-off effect based on the general kinetic and thermodynamic principles of aromatic nitration. Specifically, countercurrent operation was integrated with flow chemistry to transform the conventional single-stage co-current mode into a two-stage countercurrent mode. A microreaction system was developed featuring co-current intra-stage, countercurrent inter-stage, which enhanced overall reaction efficiency by more than 5 times. Meanwhile, the lower reactant concentrations in each stage significantly reduced the heat release rate. Under identical conditions, the interface temperature rise in the single-stage co-current system could reach 51 °C within 1 s, whereas it was only 15 °C in the two-stage countercurrent system, indicating substantially improved thermal controllability alongside efficiency enhancement. Based on the principles of flow chemistry and the advantages of microreactors, the product inhibition mechanism in aromatic nitration was revealed. Under low $H_2SO_4$ dosage, the produced $H_2O$ by the main reaction will in situ inhibit the dissolution of nitroaromatics in the aqueous phase, thereby effectively inhibiting the occurrence of over-nitration side reactions. This thermodynamic inhibition is further supported by molecular simulations and equilibrium analysis of $NO_2^+$ activity, which reveal that produced $H_2O$ simultaneously weakens nitroaromatic solvation in the acid phase and suppresses the activity of nitrating species (Supplementary Note 7).

Taking toluene nitration as an example, the two-stage countercurrent microreaction system with low $H_2SO_4$ dosage simultaneously achieves 99.9% toluene conversion and 99.8% mononitrotoluene selectivity. The spatiotemporal conversion rate was improved by 2 orders of magnitude compared with the batch nitration technology, while the selectivity of over-nitration side reactions was reduced by 1–2 orders of magnitude compared with the literature reports. The synergistic regulation of kinetics and thermodynamics was further

extended to the nitration of benzene and chlorobenzene, effectively overcoming the trade-off effect, and demonstrating the applicability of this strategy to aromatic nitration.

Beyond performance enhancement, the two-stage countercurrent system also exhibits favorable scalability. Each co-current microreactor in the two-stage countercurrent system can be independently scaled via numbering-up, enabling reproducible flow distribution and reaction performance across multiple parallel channels. Although size-up of individual microchannels reduces heat transfer efficiency, the reaction rate and selectivity can still be controlled by adjusting $H_2SO_4$ concentration and temperature according to the microreaction strategy established in this work. These features minimize reaction, separation, and storage risks, thereby enhancing the intrinsic safety of aromatic nitration processes and supporting their translation toward industrial implementation.

## Methods

### Materials

$H_2SO_4$ (98%) and $HNO_3$ (98%) were provided by Beijing Tong Guang Fine Chemicals Company. Other reagents were purchased from Aladdin Chemistry Co., Ltd. $H_2SO_4$ (<98%) was prepared by diluting $H_2SO_4$ (98%) with the deionized water. Other chemicals were used as received without any further purification. The only metallic component in contact with the mixed acid was the 316 L stainless-steel pump head of the metering pump, which was operated at room temperature. The other components in contact with the mixed acid were fabricated from acid-resistant materials.

### Reaction procedures

First, $H_2SO_4$ is diluted to the specified concentration. Next, according to the molar ratio of $HNO_3$ to aromatics and the flow rate ratio of the two phases, a specified amount of $HNO_3$ is added to the $H_2SO_4$ to obtain a mixed acid solution. The aqueous and organic phases are separately delivered using metering pumps (EMO AP-F, China). The two phases are dispersed in a microdisperser (CTFE) and reacted in a capillary microreactor (PTFE). The structure of the microdisperser is T-junction, which is consistent with our previous report[22]. The preheating coil, microdisperser, and capillary microreactor are all placed in a thermostat (CORIO CD-200F, Julabo, Germany) to maintain a constant temperature during the reaction process. Each reactor outlet is connected to a phase separator. At the outlet of the first-stage, the concentration of $HNO_3$ is nearly zero, so it can be assumed that no further reaction occurs in the phase separator. The obtained organic phase is then delivered to the second-stage for further reaction. At the outlet of the second-stage, the phase separator contains $H_2SO_4$ at a lower concentration and at a temperature significantly lower than the reaction temperature, resulting in a very slow reaction rate. Therefore, it can also be assumed that no further reaction occurs. The organic phase obtained from the second phase separator is the final product.

For the long-term stability evaluation, the reactor was operated continuously for up to 10 h, and samples were periodically collected from the reactor outlet for compositional analysis (Section 10 in Supplementary Information).

### Analysis

All samples were taken directly from the reactor outlet. The product was added to ice water to further dilute the $H_2SO_4$. After thorough mixing, methanol was added to obtain a homogeneous solution. The composition of the samples was analyzed using UPLC (Waters, ACQUITY UPLC I-CLASS System, column: PFP Column, 1.8 μm, 3 mm × 50 mm, America, mobile phase: $Q_{H2O}/Q_{MeOH}$ = 0.65/0.35, flow rate: 0.2 mL/min, injection volume: 1 μL). The content of each component was characterized using the internal standard method. GC–MS and UPLC–MS were employed for identification of by-products.

## Determination of solubility of aromatics in H$_2$SO$_4$

First, H$_2$SO$_4$ was diluted to the specified concentration. An excess of aromatic was added to the H$_2$SO$_4$, and the mixture was stirred at 800 rpm for 6 h. After centrifugation to separate the phases, the upper organic phase was decanted, and a sample was taken from the lower aqueous phase. The aromatic hydrocarbon content was measured using the UPLC.

## Data availability

The data that support the findings of this study are provided in the Source Data file, which constitutes the minimum dataset necessary to interpret and verify the results. Additional data are available in the manuscript and Supplementary Information. Source data are provided with this paper.

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

## Acknowledgements

This work was financially supported by the National Natural Science Foundation of China (Grant No. 21991104, G.L.).

## Author contributions

G.L. and J.S. conceived the idea and designed the research. G.L. supervised the project. J.S. performed the experiments. J.S., Y.P., and R.X. performed the molecular simulations. J.S. wrote and revised the manuscript, and Z.Y., T.T., K.W., Y.W., and J.D. contributed to the writing. All authors discussed the results and approved the final manuscript.

## Competing interests

The authors declare no competing interests.
