## [Transparent Peer Review file · Nature Communications]

A Countercurrent Microflow Strategy for Simultaneous High Selectivity and Conversion in Aromatic Nitration

Corresponding Author: Professor Guangsheng Luo

Version 0:

Reviewer comments:

Reviewer #1

(Remarks to the Author)

This manuscript presents a micro-reaction strategy utilizing a two-stage countercurrent flow architecture to address the long-standing trade-off between spatiotemporal conversion rate and selectivity in aromatic nitration with mixed acid. The experimental approach is systematic, and the results appear promising. However, several key aspects require further clarification and validation to firmly establish the claims, particularly regarding the proposed mechanisms and the comparative advantages of the countercurrent system. Prior to publication in Nature Communications, it is necessary to address the following comments.

Major comments:

1. The actual effect of the countercurrent mode. Whereas the conventional co-current mode is limited in its later phase by low aromatic concentration, the countercurrent mode partitions these limitations: the first stage is constrained by low HNO_3 concentration, and the second by low aromatic concentration. Therefore, the reported improvements in conversion efficiency and thermal management stem from operating each stage under more favorable conditions for one reactant, thereby raising the time-averaged reaction rate and reducing localized heat release. Although conversion improves significantly, the underlying reaction kinetics remain unchanged. The authors should clarify this distinction explicitly.
2. Kinetics versus thermodynamics. In the manuscript, the authors claim to have kinetically and thermodynamically controlled the reaction process. However, I think they haven't clearly explained certain specific concepts, which may mislead readers:
 - a. In eqs. (1) and (2) and in Section 3 ("Application of two-stage countercurrent kinetic regulation"), the authors treat the reaction as second-order and suggest that kinetics can be tuned by changing M (the HNO_3 -to-aromatic molar ratio). However, concentration changes affect the reaction rate through mass action, not the intrinsic rate constant k , which is governed by the Arrhenius equation. The authors should clarify that the observed rate enhancement is due to optimized concentration profiles, not a change in fundamental kinetics.
 - b. In Section 4 ("Suppress over-nitration side reaction via thermodynamic regulation"), H_2SO_4 concentration and temperature are presented as thermodynamic controls. While these factors influence both kinetics (via activation energy) and thermodynamics (e.g., solubility, phase behavior), the authors should explicitly distinguish their effects. For instance, H_2SO_4 acts as a catalyst and solvent, affecting both reaction rate and product distribution through physical solubility changes.
3. Unclear mechanism for selectivity. The manuscript proposes that the generated H_2O is a key intermediate during MNT formation, reducing MNT solubility and dissolution and thereby suppressing over-nitration in the aqueous phase. Since both countercurrent and normal modes generate H_2O , what specific selectivity advantage arises from the countercurrent mode? Moreover, the proposed mechanism appears speculative and lacks direct evidence.
4. Fundamental constraints are not resolved. Although the authors employ a countercurrent microflow mode, the reaction at each stage still seems to face the same constraints as the normal mode (Fig. 2a and 2b), suggesting the core trade-off remains unresolved.
5. Uniqueness of the mode remains unclear. Many variables (t , X_{ar} , M , T , WH_2SO_4 and Q) are explored under the countercurrent microflow mode, yet it remains unclear which factors are most critical in governing selectivity and conversion in their system.

Minor comments:

6. In Fig. 2h, different HNO_3 dosages are used in stages 1 and 2. Is the total nitric acid consumption the same as in the normal mode?
7. In Fig. 5e and 6b, why does the SDNT fraction decrease with increasing residence time?

8. Normalize the literature data where applicable (see Fig. 6c-6e) to indicate whether the SDNT values from the literature are comparable in terms of temperature, WH_2SO_4 , and M. If they are not, provide a reasonable performance envelope.
9. The two-stage countercurrent system requires inter-stage phase separation and recirculation of the acid phase. What are the practical challenges and potential efficiency losses (e.g., product carry-over, acid phase degradation, flow stability) associated with this separation and recirculation in a continuous system, especially at larger scales?

Reviewer #2

(Remarks to the Author)

This manuscript presents an innovative "two-stage countercurrent microreaction" strategy that successfully overcomes the classic trade-off between spatiotemporal conversion rate and selectivity in aromatic nitration through synergistic kinetic and thermodynamic control. The study is well-designed and supported by substantial experimental data, offering a groundbreaking solution to a challenge in aromatic nitration. I recommend acceptance after minor revisions. Below are detailed comments:

1. In Figure 2f, there are two orange lines. Are the HRR values of those lines measured from the first stage? This point should be expressed more clearly.
2. Although low H_2SO_4 usage is emphasized, the effect of $\text{H}_2\text{SO}_4/\text{HNO}_3$ ratio variation on selectivity is not systematically explored. Additional experiments with different HNO_3 concentrations are recommended to define the "optimal dilution window."
3. After reducing the amount of sulfuric acid, the temperature was also raised to 90°C , which might accelerate the corrosion of the microreactor. (especially in hot zones). A long-term stability test should be evaluated.
4. Does reduced over-nitration selectivity (SDNT) coincide with increased oxidative byproducts? Comprehensive byproduct analysis (e.g., HPLC-MS) is necessary to validate selectivity improvements fully.
5. By optimizing reactant concentration distribution through a two-stage countercurrent operation, this work overcomes the efficiency bottleneck imposed by intrinsic second-order reaction kinetics. If analysis of the Damköhler number (or Hatta number) under actual operating conditions could be added, it would more rigorously validate the premise that mass transfer is not rate-limiting, thereby making the argument for its kinetic optimization strategy more comprehensive and compelling.
6. The axis label should be "t (min)" instead of a slash, the current notation is confusing. Additionally, all abbreviations should be written in full the first time they appear. Please carefully review the entire text.

Reviewer #3

(Remarks to the Author)

This manuscript proposes a potentially transformative strategy for aromatic nitration by synergistically combining a novel two-stage countercurrent microflow system for kinetic regulation with a "product inhibition mechanism" for thermodynamic control. The work successfully addresses the persistent trade-off between spatiotemporal conversion rate and selectivity, demonstrating outstanding performance in toluene, benzene, and chlorobenzene nitration. However, before acceptance, the author must address the following major issues to strongly support their statement and fully clarify the reaction mechanism and practical application potential. The essential revisions primarily concern:

- 1 The authors are advised to refine the language used to describe their scientific claims throughout the manuscript, avoiding overly absolute or potentially contentious terms. Such as "firstly proposed" (line 82), "the highest" (line 22, 290, 338), etc
- 2 The authors repeatedly emphasize in the manuscript that their developed two-stage countercurrent microreaction technology improves the spatiotemporal conversion rate of aromatic nitration by "two orders of magnitude" compared to traditional batch reactors. This claim is crucial for demonstrating the extent of technological advancement in this work. However, the evidence provided in the current manuscript is insufficient to directly support this strong conclusion. Specifically, while the conclusion is stated, the manuscript lacks specific numerical values, experimental conditions, or cited sources for the spatiotemporal conversion rate of the traditional batch reactors used for comparison.
- 3 In Figures 6c, d, e, the current presentation effectively highlights the advantages in the performance outcomes but lacks a side-by-side comparison of the key operating conditions required to achieve them. To more convincingly demonstrate that the superiority of the proposed strategy stems from its intrinsic innovation rather than from sacrificing other parameters, it is recommended to provide a systematic comparison table. This table should compare, at similar conversion levels, the spatiotemporal conversion rate, selectivity, and key experimental conditions.
- 4 The manuscript only focuses on the by-product of DNT. Please provide a comprehensive product analysis spectrum (such as GC-MS) to prove that at such a high conversion rate, other competitive side reactions such as oxidation and polymerization have also been effectively controlled.
- 5 At present, the "product inhibition mechanism" is mainly explained from the perspective of macroscopic solubility differences. If molecular simulation or in-situ monitoring of the acid phase composition during the reaction process can be combined to explore the influence of water molecule addition on the hydrogen sulfate bond network, the active form of nitrifying agents, and the solvation structure of organic substances, the mechanism explanation will be more profound.
- 6 The reaction medium has strong oxidizing and corrosive properties, which poses a severe test to the material of the equipment. Please clearly identify the material and structure of the reactor, and use ICP-MS or other detection methods to

check if any substances from the reactor material have dissolved.

7 It is suggested to supplement a brief discussion on the amplification strategy of this micro-reaction system, such as the issues of distribution uniformity and temperature control faced by scale-up via numbering-up, in order to assess its industrial transformation potential.

Version 1:

Reviewer comments:

Reviewer #2

(Remarks to the Author)

The authors have carefully considered and responded to each comment from the review and have vastly improved the quality of the manuscript. So, it can be accepted as it is.

Reviewer #3

(Remarks to the Author)

I have reviewed the revised manuscript titled "A Precise Microreaction Strategy to Overcome Trade-off Effect in Aromatic Nitration via Kinetic and Thermodynamic Regulation" and the authors' detailed point-by-point response to the reviewers' comments. The authors have made substantial and thoughtful revisions that have significantly strengthened the manuscript. All major and minor concerns raised during the initial review have been adequately addressed. Therefore, I am satisfied that the manuscript in its current form meets Nature Communications.

Dear Editor:

Thanks so much for your attention and the reviewers' suggestions on our manuscript "*A Precise Microreaction Strategy to Overcome Trade-off Effect in Aromatic Nitration via Kinetic and Thermodynamic Regulation*". All comments are very helpful for improving the quality of our manuscript. We have revised the manuscript according to your kind advices and reviewers' suggestions. Enclosed please find the responses to reviewers' comments. In addition to the revised manuscript, we also uploaded one version (named Marked Revision) with highlighted changes in blue. We sincerely hope this manuscript will be finally acceptable and published on *Nature Communications*. Thank you so much for all your help and we are looking forward to hearing positive response from you soon.

Reviewer #1

1. The actual effect of the countercurrent mode. Whereas the conventional co-current mode is limited in its later phase by low aromatic concentration, the countercurrent mode partitions these limitations: the first stage is constrained by low HNO₃ concentration, and the second by low aromatic concentration. Therefore, the reported improvements in conversion efficiency and thermal management stem from operating each stage under more favorable conditions for one reactant, thereby raising the time-averaged reaction rate and reducing localized heat release. Although conversion improves significantly, the underlying reaction kinetics remain unchanged. The authors should clarify this distinction explicitly.

Thank you very much for your suggestion. The two-stage countercurrent mode does not affect the intrinsic reaction rate constant or activation energy. Instead, it redistributes the concentration gradients of the two reactants along the reactor, enabling each stage to operate under more favorable local conditions and thus enhancing overall spatiotemporal conversion rate and thermal uniformity. We have clarified this distinction explicitly in the revised manuscript in the "*Comparison of reaction kinetics between co-current and countercurrent modes*" section as follows:

“It should be noted that the two-stage countercurrent mode does not change the intrinsic reaction rate constant and activation energy, but rather redistributes the concentration gradients of the two reactants along the reactor, enabling each stage to operate under more favorable local conditions and thus enhancing overall spatiotemporal conversion rate and thermal uniformity.”

2. Kinetics versus thermodynamics. In the manuscript, the authors claim to have kinetically and thermodynamically controlled the reaction process. However, I think they haven't clearly explained certain specific concepts, which may mislead readers:

a. In eqs. (1) and (2) and in Section 3 (“Application of two-stage countercurrent kinetic regulation”), the authors treat the reaction as second-order and suggest that kinetics can be tuned by changing M (the HNO_3 -to-aromatic molar ratio). However, concentration changes affect the reaction rate through mass action, not the intrinsic rate constant k , which is governed by the Arrhenius equation. The authors should clarify that the observed rate enhancement is due to optimized concentration profiles, not a change in fundamental kinetics.

b. In Section 4 (“Suppress over-nitration side reaction via thermodynamic regulation”), H_2SO_4 concentration and temperature are presented as thermodynamic controls. While these factors influence both kinetics (via activation energy) and thermodynamics (e.g., solubility, phase behavior), the authors should explicitly distinguish their effects. For instance, H_2SO_4 acts as a catalyst and solvent, affecting both reaction rate and product distribution through physical solubility changes.

Thank you very much for your important suggestion. We have revised the relevant sections accordingly in the revised manuscript.

a. This clarification has been added to section 3, “*Application of two-stage countercurrent kinetic regulation*”:

“As mentioned above, the enhancement of the spatiotemporal conversion rate originates from the optimized reactant concentration profiles governed by the law of mass action. The molar ratio of HNO_3 to aromatic (M) serves as the key parameter that regulates the concentration distribution of the two reactants along the reactor. It also

determines the allocation of conversion between the two stages in the design of the two-stage countercurrent mode, as described by Eq. (3).”

b. This clarification has been added to section 4, “*Suppress over-nitration side reaction via thermodynamic regulation*”:

“H₂SO₄ serves as both catalyst and solvent, affecting both reaction kinetics and product distribution through physical solubility changes.”

“Its essence lies in the fact that the generated H₂O in the main reaction *in situ* reduces the solubility of the product in H₂SO₄, thereby keeping the over-nitration side reaction under thermodynamically unfavorable conditions and effectively suppressing it. The concentration and dosage of H₂SO₄ are the key parameters for achieving this thermodynamic control, which we term the “*product inhibition mechanism*” (Fig. 5h).”

3. Unclear mechanism for selectivity. The manuscript proposes that the generated H₂O is a key intermediate during MNT formation, reducing MNT solubility and dissolution and thereby suppressing over-nitration in the aqueous phase. Since both countercurrent and normal modes generate H₂O, what specific selectivity advantage arises from the countercurrent mode? Moreover, the proposed mechanism appears speculative and lacks direct evidence.

Thank you very much for your important comment. The *product inhibition mechanism*, as a thermodynamic regulation strategy, applies to both co-current and countercurrent systems. However, it is more effectively manifested under the countercurrent mode. As shown in Fig. 6b, even under co-current mode, the selectivity of the over-nitration side reaction is significantly lower than those reported in the literature, confirming that the product inhibition mechanism is generally valid.

In the co-current mode, HNO₃ and aromatics are simultaneously present at high concentrations at the reactor inlet, leading to a rapid reaction rate and intense heat release. The resulting interfacial temperature rise increases the solubility of MNT in H₂SO₄ (Figs. 2g and 5g). In addition, since the activation energy of the over-nitration reaction is higher than that of mononitration, elevated temperatures further promote its occurrence. Moreover, as shown in Fig. 6a, a longer residence time is required to reach

high conversion in the co-current system, which enhances the probability of over-nitration as the second step of the consecutive reactions.

In contrast, the two-stage countercurrent mode significantly reduces the interfacial temperature rise (Fig. 2g) and overall residence time (Fig. 2e). These effects suppress the conditions favorable for over-nitration and further strengthen the inhibition effect induced by the generated H₂O, resulting in higher selectivity.

In addition, we quantified the solubility evolution of aromatics and nitroaromatics as a function of H₂SO₄ concentration (Fig. 5g), and determined the dynamic variation of H₂SO₄ concentration with residence time (Fig. 5c,f). Molecular simulations further reveal that the solvation of nitroaromatics in H₂SO₄ originates from directional hydrogen bonding between the nitro group (-NO₂) and the -OH groups of H₂SO₄. This provides a molecular-level foundation for the product inhibition mechanism and is fully consistent with the experimentally observed sharp decrease in nitroaromatic solubility as the H₂SO₄ concentration decreases. These results demonstrate that the in-situ generated H₂O weakens the solvating ability of the acid phase toward nitroaromatics, thereby suppressing their diffusion into the acid phase. Moreover, we calculated the equilibrium concentration of nitronium ion (NO₂⁺) under different temperatures and H₂SO₄ concentrations, and the results show that generated H₂O substantially reduces the NO₂⁺ activity. This indicates that the droplet interface becomes a region where the over-nitration is simultaneously thermodynamically and kinetically disfavored.

Therefore, by lowering the H₂SO₄ fraction in the mixed acid, we intensify the dilution effect induced by H₂O and effectively suppress the over-nitration. The experimental observations and molecular simulations provide direct evidence supporting the proposed product inhibition mechanism.

4. Fundamental constraints are not resolved. Although the authors employ a countercurrent microflow mode, the reaction at each stage still seems to face the same constraints as the normal mode (Fig. 2a and 2b), suggesting the core trade-off remains unresolved.

Thank you very much for your important comment. The trade-off effect in

aromatic nitration refers to the intrinsic difficulty of simultaneously achieving high spatiotemporal conversion rate and high selectivity. Its essence is that an increase in the reaction rate is inevitably accompanied by an increase in the heat release rate, leading to a high interfacial temperature rise (ΔT). Since the activation energy of the over-nitration side reaction is higher than that of mononitration, the elevated temperatures accelerate over-nitration and decrease the selectivity of the mononitration. The constraint still exists in single-stage co-current reactors. Although high selectivity can be achieved through the product inhibition mechanism in single-stage co-current reactors, it is obtained at the cost of low spatiotemporal conversion rate.

In contrast, the two-stage countercurrent mode moderates the reaction intensity of each stage while simultaneously enhancing the overall spatiotemporal conversion rate. By combining “countercurrent inter-stage” with product inhibition mechanism, this design effectively overcomes the trade-off limitation of single-stage reactor.

5. Uniqueness of the mode remains unclear. Many variables (t , X_{ar} , M , T , $w_{H_2SO_4}$ and Q) are explored under the countercurrent microflow mode, yet it remains unclear which factors are most critical in governing selectivity and conversion in their system.

Thank you very much for your kind suggestion. The uniqueness of the two-stage countercurrent mode lies in dividing a single-stage co-current reactor into a countercurrent flow between two co-current reactors, leading to the improvement in spatiotemporal conversion rate. Therefore, a rational segmentation design is the core of the two-stage countercurrent strategy. As shown in Eq. (3), the molar ratio of HNO_3 to aromatic (M) is the key parameter for designing the conversion distribution in the two-stage countercurrent mode. It defines the spatial concentration gradients of the two reactants within each stage, thereby influencing both the reaction rate and the heat release rate. The mass fraction of H_2SO_4 ($w_{H_2SO_4}$) determines the solubility of aromatic and nitroaromatics in H_2SO_4 , while the H_2SO_4 dosage determines whether the product inhibition mechanism can function. When M is fixed, the H_2SO_4 dosage is governed by the molar ratio of H_2SO_4 to HNO_3 ($n_{H_2SO_4}/n_{HNO_3}$). Therefore, both $w_{H_2SO_4}$ and $n_{H_2SO_4}/n_{HNO_3}$ together determine the selectivity. Overall, M , $w_{H_2SO_4}$, and $n_{H_2SO_4}/n_{HNO_3}$

are the most critical parameters in governing selectivity and conversion in the two-stage countercurrent mode.

Minor comments:

6. In Fig. 2h, different HNO₃ dosages are used in stages 1 and 2. Is the total nitric acid consumption the same as in the normal mode?

Thank you very much for your question. As shown in Fig. 4a, the total input of HNO₃ is 1.01 molar equivalents ($M = 1$). In the two-stage countercurrent configuration, HNO₃ feed enters from the inlet of the second-stage reactor and, after partial consumption, the remaining HNO₃ is recycled back to the first-stage reactor. No additional HNO₃ is added. Therefore, the overall HNO₃ input and consumption are the same as that in the normal mode.

7. In Fig. 5e and 6b, why does the SDNT fraction decrease with increasing residence time?

Thank you very much for your question. The observed decrease in S_{DNT} with increasing residence time arises from the dilution effect of H₂SO₄ by the generated H₂O, which strengthens the product inhibition mechanism. In the early stage of the reaction, the reaction rate and heat release rate are high. Although the generated H₂O lowers the H₂SO₄ concentration, the elevated interfacial temperature simultaneously increases the solubility of mononitrotoluene (MNT) in H₂SO₄, leading to the formation of a small amount of DNT. As the reaction proceeds, the reaction rate decreases, leading to a lower interfacial temperature rise and a lower H₂SO₄ concentration. Under these conditions, the over-nitration side reaction becomes thermodynamically unfavorable, and its rate is much lower than that of the main nitration reaction. Consequently, the fraction of DNT continuously decreases with increasing residence time.

8. Normalize the literature data where applicable (see Fig. 6c-6e) to indicate whether the SDNT values from the literature are comparable in terms of temperature, WH₂SO₄, and M. If they are not, provide a reasonable performance envelope.

Thank you very much for your kind suggestion. We have added the detailed reaction conditions corresponding to the literature data (Figs. 6c-e) to the revised manuscript (Table S1) to clarify their comparability. The literature results are all obtained in single-stage co-current microreactors. For comparison, we additionally conducted single-stage co-current experiments under the same operating conditions as the two-stage countercurrent process.

As shown in Table S1, with only a 1% excess of HNO₃, the single-stage co-current mode achieved 69% toluene conversion at the same residence time, whereas the two-stage countercurrent mode reached 99.9%. This clearly demonstrates the advantage of the countercurrent configuration in enhancing the spatiotemporal conversion rate. Moreover, the mononitration selectivity achieved in both systems was significantly higher than the values reported in the literature, confirming that the product inhibition mechanism effectively improves selectivity.

Table S1. Comparison of reaction performance of toluene nitration.

Authors (Year)	Reaction mode	V (mL)	w _{H₂SO_{4,0}} (%)	M	T (°C)	t (min)	x _T (%)	S _{MNT} (%)	STCR (mol/L·min)
Panke et al. (2003) ¹	Batch	25	62	1.5	65	140	100	94.6	1.3×10 ⁻⁵
Burns et al. (2002) ²		0.0238	80	HNO ₃ excessive	RT*	0.2	80%	63.3	6.3×10 ⁻³
Panke et al. (2003) ¹		1.2	62	1.5	65	15.0	98.8	89.5	3.1×10 ⁻⁴
Fu et al. (2022) ³		132	—	1.2	45	2.2	86	95	1.7×10 ⁻⁴
Yang et al. (2022) ⁴		10	58	1.3	49	2.8	85	91.3	1.7×10 ⁻³
Han et al. (2024) ⁵	Continuous	6.74	80	1.4	70	1.2	100	91.5	6.3×10 ⁻³
Han et al. (2024) ⁵		6.65	79	1.3	80	1.0	100	89.9	7.5×10 ⁻³
Han et al. (2024) ⁵		6.82	78	1.4	70	1.2	99.1	90.6	6.2×10 ⁻³
Single-stage co-current (This work)		4	71.5	1.01	90	3.9	69	99.1	0.8×10 ⁻³
Two-stage countercurrent (This work)		4	71.5	1.01	90	3.9	99.9	99.8	1.2×10 ⁻³

RT* means room temperature.

9. The two-stage countercurrent system requires inter-stage phase separation and recirculation of the acid phase. What are the practical challenges and potential efficiency losses (e.g., product carry-over, acid phase degradation, flow stability) associated with this separation and recirculation in a continuous system, especially at larger scales?

Thank you very much for your important comment. Phase separation and inter-stage countercurrent flow are indeed the core features that distinguish the proposed two-stage countercurrent mode. In our microreaction nitration strategy, the initial H_2SO_4 concentration is selected based on the low solubilities of both nitroaromatics and aromatics in H_2SO_4 . As nitration proceeds, the in-situ generated water further dilutes H_2SO_4 , and reduces the solubility of the organic phase in the aqueous phase. As a result, the amount of organic phase entrained in the aqueous phase after reaction is extremely low. Meanwhile, the two-stage countercurrent mode achieves high aromatic conversion without requiring large excess of HNO_3 . Therefore, the residual HNO_3 in the aqueous phase is also low. Furthermore, the content of by-products is extremely low, so the acid phase does not undergo oxidative degradation or coking due to impurity accumulation. These characteristics greatly facilitate the H_2SO_4 reconcentration and recycling.

Regarding phase separation, the mixed acid acts as the dispersed phase to enable the product inhibition mechanism in this work. The interfacial tension between mixed acid and aromatics is relatively high ($> 15 \text{ mN/m}$). It will increase as the water content increases (for reference, the interfacial tension between water and toluene is 35 mN/m). This favors rapid coalescence of mixed acid droplets. In addition, a large density difference exists between the two phases ($1.5 - 1.7 \text{ g/mL}$ for mixed acid and $0.9 - 1.1 \text{ g/mL}$ for aromatics), enabling efficient gravity-based separation. This property also ensures stable inter-stage countercurrent flow. In the microreactor, uniform channel dimensions ensure stable and robust flow behavior.

We believe that heat management is the most critical challenge in scaling-up of the two-stage countercurrent system. As shown in Fig. 2g, although the countercurrent

mode significantly reduces interfacial temperature rise, the initial reaction stage still exhibits relatively high heat release rates. In this work, PTFE microreactors were selected to maintain the organic phase as the continuous phase. Although PTFE material has moderate thermal conductivity, the small liquid holdup and high specific surface area of microreactors provide effective heat transfer ability. However, at large industrial throughputs, heat-management requirements will become more stringent, and reactor design must account for enhanced heat transfer capability.

A summary of these considerations has been added to the “Discussion” section of the revised manuscript.

Reviewer #2

1. In Figure 2f, there are two orange lines. Are the HRR values of those lines measured from the first stage? This point should be expressed more clearly.

Thank you very much for your kind suggestion. The two orange lines in Fig. 2f correspond to the heat release rates (*HRR*) of the first-stage and second-stage microreactors in the two-stage countercurrent system, respectively. Since the two microreactors are controlled independently, the two lines represent the *HRR* profiles along the flow direction (from inlet to outlet) for each stage. We have revised Fig. 2f in the revised manuscript.

2. Although low H₂SO₄ usage is emphasized, the effect of H₂SO₄/HNO₃ ratio variation on selectivity is not systematically explored. Additional experiments with different HNO₃ concentrations are recommended to define the "optimal dilution window."

Thank you very much for your kind suggestion. We have supplemented the analysis of the effect of the H₂SO₄/HNO₃ molar ratio on the aromatic nitration performance, as shown in Fig. S1. The tested range spans from fuming HNO₃ ($n_{\text{H}_2\text{SO}_4}/n_{\text{HNO}_3} = 0$) to the conventional mixed acid compositions used in conventional aromatic nitration technologies ($n_{\text{H}_2\text{SO}_4}/n_{\text{HNO}_3} > 5$). The results show that when fuming HNO₃ is used as the nitrating agent ($n_{\text{H}_2\text{SO}_4}/n_{\text{HNO}_3} = 0$), the reaction rate is significantly lower than that obtained under H₂SO₄ catalyzed conditions, while the extent of over-nitration increases markedly. This is because with fuming HNO₃ as the nitrating agent, the high HNO₃ concentration leads to an intense reaction at the initial stage of reaction and a substantial interfacial temperature rise. Although both toluene and MNT are insoluble in fuming HNO₃, the elevated interfacial temperature promotes the over-nitration reaction kinetically. As the reaction proceeds, H₂O generated by mononitration rapidly dilutes HNO₃, suppressing NO₂⁺ formation and resulting in a low overall conversion. When H₂SO₄ is added, it is used as both a catalyst and a dehydrating agent, effectively absorbing the generated water and maintaining a higher reaction rate compared to using fuming HNO₃ alone. However, when the H₂SO₄ dosage is low ($0 <$

$n_{\text{H}_2\text{SO}_4}/n_{\text{HNO}_3} < 0.7$), H_2O generated still dilutes H_2SO_4 to a level insufficient for NO_2^+ generation. Conversely, when the H_2SO_4 dosage is high ($n_{\text{H}_2\text{SO}_4}/n_{\text{HNO}_3} > 1.5$), the product inhibition mechanism cannot be realized. Considering both reaction rate and selectivity, our results indicate that maintaining $n_{\text{H}_2\text{SO}_4}/n_{\text{HNO}_3}$ within 0.7 – 1.5 provides an optimal balance, enabling both high conversion and high mononitration selectivity. This corresponds to an optimal HNO_3 dilution window of 40–59 mol%.

These results have been added to the revised manuscript and Supplementary Information.

Fig. S1. Effect of molar ratio of H_2SO_4 to HNO_3 ($n_{\text{H}_2\text{SO}_4}/n_{\text{HNO}_3}$) on reaction performance. a. Effect of $n_{\text{H}_2\text{SO}_4}/n_{\text{HNO}_3}$ on toluene nitration. b. Effect of $n_{\text{H}_2\text{SO}_4}/n_{\text{HNO}_3}$ on DNT selectivity.

3. After reducing the amount of sulfuric acid, the temperature was also raised to $90\text{ }^\circ\text{C}$, which might accelerate the corrosion of the microreactor. (Especially in hot zones). A long-term stability test should be evaluated.

Thank you very much for your important comment. In this work, to fully utilize the product inhibition mechanism, the mixed acid phase was dispersed as droplets, with the organic phase serving as the continuous phase. Therefore, both the micro-disperser and the microreactor were fabricated from oleophilic and acid-resistant fluoropolymers. The micro-disperser was made of CTFE, and the microreactor was made of PTFE. To evaluate the material stability under operating conditions, we analyzed the metal ion content in the aqueous phase using ICP-OES, as shown in Table S2. The results indicate

that the reaction system remained chemically stable despite the strongly oxidizing and corrosive characteristics of aromatic nitration system. The metering pump used to deliver the mixed acid always operated at room temperature, and the presence of HNO₃ is known to reduce the corrosiveness of the mixture toward stainless steel. We also conducted a 10-hour continuous operation test, during which samples were collected every hour for analysis (Fig. S2). Throughout the operation, both the toluene conversion and the over-nitration selectivity remained stable within the experimental error range, demonstrating the robustness of the microreaction system.

These results confirm that the microreactor system maintained stable under the tested conditions. The corresponding results and discussion have been incorporated into the revised manuscript and Supplementary Information.

Fig. S2. Long-term stability test. a. Toluene conversion as a function of time-on-stream. b. DNT selectivity as a function of time-on-stream.

Table S2. Concentration of Metal Elements in Different Samples

Sample	Fe (mg/L)	Cr (mg/L)	Ni (mg/L)	Cu (mg/L)	Al (mg/L)
1*	0.04 ± 0.01	0	0	0	0
2*	0.43 ± 0.02	0.05 ± 0.01	0	2.15 ± 0.04	3.29 ± 0.05
3*	0.50 ± 0.04	0.06 ± 0.02	0	1.82 ± 0.11	3.37 ± 0.03

* Sample 1 is a mixed acid raw material. Sample 2 is a mixed acid raw material pumped to the reactor

inlet. Sample 3 is the aqueous solution after the reaction.

4. Does reduced over-nitration selectivity (SDNT) coincide with increased oxidative byproducts? Comprehensive byproduct analysis (e.g., HPLC-MS) is necessary to validate selectivity improvements fully.

Thank you very much for your kind suggestion. To verify whether reduced over-nitration is accompanied by increased oxidative byproducts, we performed a comprehensive byproduct analysis using UPLC-MS and GC-MS, as shown in Figs. S3 and S4.

Possible byproducts in the toluene nitration system include:

Over-nitration byproducts: 2,6-dinitrotoluene(2,6-DNT), 2,4-dinitrotoluene(2,4-DNT), 2,5-dinitrotoluene(2,5-DNT), 3,4-dinitrotoluene(3,4-DNT).

Oxidation and polymerization by-products: benzaldehyde, benzyl alcohol, *p*-nitrobenzaldehyde, 2,4-dinitro-*o*-cresol, 2-methylbenzophenone, 3-methylbenzophenone and 4-methylbenzophenone.

The UPLC-MS and GC-MS results were consistent, confirming the reliability of product analyzation. GC-MS was used to quantify the product distribution. At the first-stage outlet (Fig. S4a), four main peaks were observed, corresponding to toluene, *o*-MNT, *m*-MNT, and *p*-MNT. The total content of over-nitration products was <0.05%, while oxidation/polymerization products were less than 0.01%. At the second-stage outlet (Fig. S4b), only *o*-MNT, *m*-MNT, and *p*-MNT were detected. The residual toluene was <0.1%, and the level of over-nitration products remained very low (<0.2%). Consistently, oxidation and polymerization byproducts were less than 0.01%.

These results clearly demonstrate that our proposed microreaction strategy suppresses over-nitration side reactions without increasing the occurrence of oxidation or polymerization reactions.

The full chromatographic spectra, mass-fragmentation assignments, and detailed discussion have been added to the Supplementary Information and incorporated into the revised manuscript.

Fig. S3. UPLC-MS spectra of products from the toluene nitration system under two-stage countercurrent mode. a, product from the organic phase of the first-stage microreactor outlet. b, product from the organic phase of the second-stage microreactor outlet.

Fig. S4. GC-MS spectra of products from the toluene nitration system under two-stage countercurrent mode. a, product from the organic phase of the first-stage microreactor outlet. b, product from the organic phase of the second-stage microreactor outlet.

5. By optimizing reactant concentration distribution through a two-stage countercurrent operation, this work overcomes the efficiency bottleneck imposed by intrinsic second-order reaction kinetics. If analysis of the Damköhler number (or Hatta number) under actual operating conditions could be added, it would more rigorously validate the premise that mass transfer is not rate-limiting, thereby making the argument for its

kinetic optimization strategy more comprehensive and compelling.

Thank you very much for your kind suggestion. We evaluated the relative contributions of reaction kinetics and mass transfer under the actual operating conditions by calculating the Hatta number (Ha), as shown in Fig. S5. The calculation method follows our previously reported approach. The results show that, in both stages of the microreactor, the Hatta number remains below 0.3 and gradually decreases with increasing residence time. This confirms that throughout the entire reaction process, mass transfer is significantly faster than the intrinsic reaction rate. Therefore, the overall reaction rate is kinetically controlled rather than mass-transfer limited.

As the reaction proceeds, the concentrations of both reactants and the catalytic acid decrease, further lowering the intrinsic reaction rate and thus reducing the Hatta number. These results establish that the enhancement in spatiotemporal conversion rate achieved by the two-stage countercurrent operation arises from the redistribution of reactant concentration gradients, rather than from any improvement in mass transfer. This proves the effectiveness of the mechanism of the kinetic control strategy proposed in this work.

Fig. S5. Variation of Hatta number (Ha) with residence time (t) in two countercurrent reaction process. a. Changes in Ha with t in the first-stage microreactor. b. Changes in Ha with t in the second-stage microreactor.

6. The axis label should be "t (min)" instead of a slash, the current notation is confusing.

Additionally, all abbreviations should be written in full the first time they appear. Please carefully review the entire text.

Thank you very much for your kind reminder. We have revised the axis labels in all figures by replacing the previous slash with the standard format. In addition, we carefully checked the entire manuscript, figure captions, and Supplementary Information, and ensured that all abbreviations are written in full at their first appearance and remain consistent throughout the text.

Reviewer #3

1 The authors are advised to refine the language used to describe their scientific claims throughout the manuscript, avoiding overly absolute or potentially contentious terms. Such as " firstly proposed "(line 82), "the highest" (line 22, 290,338), etc

Thank you very much for your reminder. We have carefully re-examined the full manuscript and revised the statements that may appear overly absolute or potentially contentious.

2 The authors repeatedly emphasize in the manuscript that their developed two-stage countercurrent microreaction technology improves the spatiotemporal conversion rate of aromatic nitration by "two orders of magnitude" compared to traditional batch reactors. This claim is crucial for demonstrating the extent of technological advancement in this work. However, the evidence provided in the current manuscript is insufficient to directly support this strong conclusion. Specifically, while the conclusion is stated, the manuscript lacks specific numerical values, experimental conditions, or cited sources for the spatiotemporal conversion rate of the traditional batch reactors used for comparison.

Thank you very much for your reminder. We have added the reactor reaction conditions, space-time conversion rate, and references to the revised manuscript, as shown in Table S1. The relevant calculation details have been added to the Supplementary Information.

Table S1. Comparison of reaction performance of toluene nitration.

Authors (Year)	Reaction mode	V (mL)	$w_{\text{H}_2\text{SO}_4,0}$ (%)	M	T (°C)	t (min)	x_T (%)	S_{MNT} (%)	$STCR$ (mol/L·min)
Panke et al. (2003) ¹	Batch	25	62	1.5	65	140	100	94.6	1.3×10^{-5}
Burns et al. (2002) ²	Continuous	0.0238	80	HNO_3 excessive	RT*	0.2	80%	63.3	6.3×10^{-3}
Panke et al. (2003) ¹		1.2	62	1.5	65	15.0	98.8	89.5	3.1×10^{-4}
Fu et al. (2022) ³		132	—	1.2	45	2.2	86	95	1.7×10^{-4}

Yang et al. (2022) ⁴	10	58	1.3	49	2.8	85	91.3	1.7×10^{-3}
Han et al. (2024) ⁵	6.74	80	1.4	70	1.2	100	91.5	6.3×10^{-3}
Han et al. (2024) ⁵	6.65	79	1.3	80	1.0	100	89.9	7.5×10^{-3}
Han et al. (2024) ⁵	6.82	78	1.4	70	1.2	99.1	90.6	6.2×10^{-3}
Single-stage co-current (This work)	4	71.5	1.01	90	3.9	69	99.1	0.8×10^{-3}
Two-stage countercurrent (This work)	4	71.5	1.01	90	3.9	99.9	99.8	1.2×10^{-3}

RT* means room temperature.

3. In Figures 6c, d, e, the current presentation effectively highlights the advantages in the performance outcomes but lacks a side-by-side comparison of the key operating conditions required to achieve them. To more convincingly demonstrate that the superiority of the proposed strategy stems from its intrinsic innovation rather than from sacrificing other parameters, it is recommended to provide a systematic comparison table. This table should compare, at similar conversion levels, the spatiotemporal conversion rate, selectivity, and key experimental conditions.

Thank you very much for your kind reminder. We have added a detail comparison table to the Supplementary Information (Table S1). This table compared the reaction parameters of the two-stage countercurrent process with that of both the single-stage co-current process used in this work and representative literature reports.

All nitration processes reported in the literature were carried out in single-stage co-current microreactors, differing only in reactor structure or reaction conditions. Compared with these studies, our work employs lower H₂SO₄ concentration, smaller excess of HNO₃, and higher operating temperature. Since decreasing H₂SO₄ concentration significantly reduces the reaction rate, a direct comparison under identical conditions is necessary to evaluate reaction performance. Therefore, we conducted nitration experiments using a conventional single-stage co-current microreactor under the same reaction conditions as those used in the two-stage

countercurrent mode. The results show that, within the same residence time, the single-stage co-current system achieves only 69% conversion of toluene. At this point, achieving higher conversion would require several-fold longer residence times. In contrast, the two-stage countercurrent mode reaches 99.9% conversion under the same total residence time, demonstrating its intrinsic advantage in enhancing spatiotemporal conversion efficiency. Furthermore, the H₂SO₄ dosage used in both the single-stage co-current and two-stage countercurrent modes in this work is substantially lower than those reported in the literature. Owing to the product inhibition mechanism, over-nitration remains at a very low level regardless of the flow mode. However, the two-stage countercurrent mode provides additional benefits by reducing the heat release rate and interfacial temperature rise in each stage, thereby achieving even lower over-nitration selectivity.

In summary, these results clearly demonstrate that the superiority of the proposed two-stage countercurrent strategy arises from its intrinsic innovation, rather than from sacrificing other reaction parameters.

4. The manuscript only focuses on the by-product of DNT. Please provide a comprehensive product analysis spectrum (such as GC-MS) to prove that at such a high conversion rate, other competitive side reactions such as oxidation and polymerization have also been effectively controlled.

Thank you very much for your kind suggestion. To verify the suppression of side reactions in aromatic nitration, a comprehensive product analysis was performed using the toluene nitration system as a representative sample. First, we identified all potential products and byproducts in toluene nitration system under batch conditions. As shown in Fig. S6, GC-MS confirmed the presence of the mononitration products (*o*-, *m*-, and *p*-nitrotoluene) as well as possible side products, including:

Over-nitration byproducts: 2,6-dinitrotoluene(2,6-DNT), 2,4-dinitrotoluene(2,4-DNT), 2,5-dinitrotoluene(2,5-DNT), 3,4-dinitrotoluene(3,4-DNT)

Oxidation and polymerization by-products: benzaldehyde, benzyl alcohol, *p*-nitrobenzaldehyde, 2,4-dinitro-*o*-cresol, 2-methylbenzophenone, 3-

methylbenzophenone and 4-methylbenzophenone

Based on this profiling, we further conducted full product analysis under the two-stage countercurrent reaction conditions *via* GC–MS (Fig. S7). At the first-stage outlet (Fig. S7a), four main peaks corresponding only to toluene, *o*-MNT, *m*-MNT, and *p*-MNT were detected. The content of over-nitration byproducts is less than 0.05%, and no oxidation or polymerization byproducts were detected above the instrumental detection limit (< 0.01%). At the second-stage outlet (Fig. S7b), only three main peaks corresponding to *o*-MNT, *m*-MNT, and *p*-MNT were detected. The residual toluene content was less than 0.1%, the content of over-nitration products remained very low (< 0.2%), and similarly no oxidation or polymerization products were detected above the detection limit (<0.01%).

These results clearly confirm that even at an overall conversion of 99.9%, over-nitration, oxidation and polymerization side reactions are effectively suppressed in the two-stage countercurrent microreaction system. The corresponding GC–MS spectra and discussion have been added to the Supplementary Information, and a summary statement has been added into the revised manuscript.

Fig. S6. GC-MS spectra of products from the toluene nitration system under batch conditions.

Fig. S7. GC-MS spectra of products from the toluene nitration system under two-stage countercurrent mode. a, product from the organic phase of the first-stage microreactor outlet. b, product from the organic phase of the second-stage microreactor outlet.

5. At present, the "product inhibition mechanism" is mainly explained from the perspective of macroscopic solubility differences. If molecular simulation or in-situ monitoring of the acid phase composition during the reaction process can be combined to explore the influence of water molecule addition on the hydrogen sulfate bond network, the active form of nitrifying agents, and the solvation structure of organic

substances, the mechanism explanation will be more profound.

Thank you very much for your kind suggestion. We further performed molecular simulations to investigate the solvation structure and hydrogen bonding interactions of nitroaromatic molecules in H_2SO_4 and H_2O . The results are summarized below.

Fig. S8. Molecular simulations of solvation and hydrogen bonding interactions of *p*-nitrotoluene in (a) H_2SO_4 and (b) H_2O .

p-nitrotoluene was used as a representative example. As shown in Fig. S8a, when *p*-nitrotoluene is solvated in H_2SO_4 , its nitro group ($-\text{NO}_2$) forms strong and directional hydrogen bonds with the $-\text{OH}$ groups of H_2SO_4 . Other sites on the aromatic ring show negligible interactions. This indicates that solvation of nitroaromatics in H_2SO_4 is dominated by nitro- H_2SO_4 hydrogen bond, consistent with their high solubility in H_2SO_4 . When *p*-nitrotoluene is solvated in H_2O , $-\text{NO}_2$ still forms hydrogen bonds with H_2O molecules (Fig. S8b). However, the hydrogen bond distances are significantly longer and the interactions are weaker than those in H_2SO_4 . Therefore, nitroaromatics exhibit much lower solubility in H_2O than in H_2SO_4 .

Table S3 summarizes the evolution of hydrogen bond networks at different water contents. In anhydrous H_2SO_4 , the hydrogen bond network is dominated by H_2SO_4 - H_2SO_4 interactions, with 200 O-H bonds and 53 O-H \cdots O hydrogen bonds. As water is introduced, the number of O-H groups originating from H_2SO_4 decreases (from 200 to 37 at 75% water), while H_2O -derived O-H groups increase correspondingly (from 0 to 163). Thus, the hydrogen bond network gradually transitions from a H_2SO_4 -dominated structure to a H_2O -dominated structure.

The total number of hydrogen bond increases from 53 to 76 as H_2O molar fraction rises from 0% to 75%, indicating that the network becomes more extensively hydrogen bridged rather than being disrupted. However, this network is chemically less favorable

for nitroaromatic solvation. Because solvation of nitroaromatics depends specifically on nitro- H_2SO_4 hydrogen bonds (as shown in Fig. S8). As water increases, H_2SO_4 molecular participates less in the hydrogen bond network, so the energy of breaking the existing hydrogen bond network and solvating nitroaromatics becomes higher. Therefore, nitroaromatic molecules are thermodynamically disfavored from dissolving into the aqueous phase. This microscopic insight is fully consistent with the macroscopic solubility trends we observed experimentally. This is also a common rule in aromatic nitration systems.

Table S3. Evolution of hydrogen bond network in H_2SO_4 - H_2O mixtures at different H_2O molar fractions.

H_2O molar fraction (%)	Number of O-H bonds in H_2SO_4	Number of O-H bonds in H_2O	Total O-H...O Hydrogen Bonds
0	200	0	53
25	143	57	58
50	82	118	73
75	37	163	76
100	0	200	74

We further calculated the concentration of nitronium ion (NO_2^+) at different temperature and H_2SO_4 concentration (Fig. S9). Since NO_2^+ can only exist in the aqueous phase, aromatic nitration must occur either within the acid phase or at the phase interface. The calculations show that the generation of water significantly decreases the equilibrium concentration of NO_2^+ . In other words, in-situ water locally dilutes H_2SO_4 at the interface, reducing the activity of NO_2^+ .

This effect thermodynamically disfavors the dissolving of nitroaromatics into the acid phase while also kinetically suppressing the rate of over-nitration. As a result, the interfacial region becomes a zone of simultaneous thermodynamic and kinetic suppression of over-nitration.

In summary, these simulations demonstrate that the generated H_2O during the main reaction simultaneously weakens interfacial solvation of nitroaromatics, preventing their diffusion into the acid bulk, and suppresses the activity of NO_2^+ , decreasing the

driving force for over-nitration. These provide a direct molecular support for the proposed product inhibition mechanism. The discussion has been added into the Supplementary Information and revised manuscript.

Fig. S9. Effect of temperature and H_2SO_4 concentration on NO_2^+ concentration.

6. The reaction medium has strong oxidizing and corrosive properties, which poses a severe test to the material of the equipment. Please clearly identify the material and structure of the reactor, and use ICP-MS or other detection methods to check if any substances from the reactor material have dissolved.

Thank you very much for your important comment. In this work, to fully utilize the product inhibition mechanism, the mixed acid was dispersed as droplets while the organic phase served as the continuous phase. Therefore, both the microdispenser and the microreactor were fabricated from oleophilic and acid-resistant fluoropolymers. The micro-disperser was made of chlorotrifluoroethylene (CTFE), the microreactor of polytetrafluoroethylene (PTFE), and the phase separator of glass. All of the materials exhibit excellent resistance to the mixed acid of HNO_3 and H_2SO_4 . In our experimental setup, the only metallic component in direct contact with the mixed acid is the 316L stainless-steel pump head of the metering pump. All other parts are metal-free.

To assess whether any corrosion or dissolution occurred under operating conditions, we analyzed the aqueous phase composition at different process locations

using ICP-OES (Inductively Coupled Plasma Optical Emission Spectroscopy). We monitored characteristic elements associated with stainless-steel corrosion, including Fe, Cr, Ni, Cu, and Al. As shown in Table S3, the analysis indicates trace levels of metal ions derived from the minor leaching from the 316L stainless-steel pump head. More importantly, no signals indicating corrosion and degradation of the micro-disperser, microreactor or phase separator were observed.

These results demonstrate that the reaction system remain stable under the strongly oxidizing and corrosive conditions of aromatic nitration. The detailed material specifications and ICP–OES analysis have been added to the Methods and Supplementary Information sections in the revised manuscript.

Table S2. Concentration of Metal Elements in Different Samples

Sample	Fe (mg/L)	Cr (mg/L)	Ni (mg/L)	Cu (mg/L)	Al (mg/L)
1*	0.04 ± 0.01	0	0	0	0
2*	0.43 ± 0.02	0.05 ± 0.01	0	2.15 ± 0.04	3.29 ± 0.05
3*	0.50 ± 0.04	0.06 ± 0.02	0	1.82 ± 0.11	3.37 ± 0.03

* Sample 1 is a mixed acid raw material. Sample 2 is a mixed acid raw material pumped to the reactor inlet. Sample 3 is the aqueous solution after the reaction.

7. It is suggested to supplement a brief discussion on the amplification strategy of this micro-reaction system, such as the issues of distribution uniformity and temperature control faced by scale-up via numbering-up, in order to assess its industrial transformation potential.

Thank you very much for your kind suggestion. The two-stage countercurrent microflow mode is characterized by “co-current intra-stage, and countercurrent inter-stage”. Therefore, each co-current microreactor can be independently scaled up *via* numbering-up, following common microreactor scale-up strategies. In microreactor systems, micro-dispersion, flow, and mass/heat transfer are strongly governed by channel geometry and characteristic dimensions. To date, the micro-disperser, microreactor, phase separator, and material distributor can be easily developed into

modular units. By increasing the number of modules and operating them in parallel, uniform flow distribution and reproducible reaction performance can be reliably maintained. In the aspect of thermal management, the high specific surface area of microreactors provides excellent heat transfer capability. More importantly, the two-stage countercurrent mode reduces the heat release rate within each stage microreactor, thereby lowering the thermal load and significantly simplifying temperature control during scale-up. Therefore, even under parallel numbering-up, efficient heat removal can be sustained. These features minimize the risk of localized hot spots that arise in conventional aromatic nitration processes. In summary, this modular design makes the two-stage countercurrent reaction mode highly scalable to industrial scale.

A summary of these scale-up considerations has been added to the “Discussion” section of the revised manuscript.

Reference

1. Panke G, Schwalbe T, Stirner W, Taghavi-Moghadam S, Wille G. A Practical Approach of Continuous Processing to High Energetic Nitration Reactions in Microreactors. *Synthesis* **2003**, 2827-2830 (2003).
2. Burns JR, Ramshaw C. A Microreactor for the Nitration of Benzene and Toluene. *Chemical Engineering Communications* **189**, 1611-1628 (2002).
3. Fu G, Ni L, Wei D, Jiang J, Chen Z, Pan Y. Scale-up and safety of toluene nitration in a meso-scale flow reactor. *Process Safety and Environmental Protection* **160**, 385-396 (2022).
4. Yang A, *et al.* Experimental investigation of mononitrotoluene preparation in a continuous-flow microreactor. *Research on Chemical Intermediates* **48**, 4373-4390 (2022).
5. Han B, Chen Y, Zou H, Yu G, Sheng C, Wang G. Study on characteristics of toluene/chlorobenzene nitration in different microreactors. *Chemical Engineering Research and Design* **205**, 343-353 (2024).